# In Vitro Analysis of LPS-Induced miRNA Differences in Bovine Endometrial Cells and Study of Related Pathways

**DOI:** 10.3390/ani14233367

**Published:** 2024-11-22

**Authors:** Xinmiao Li, Zhihao Zhang, Xiangnan Wang, Ligang Lu, Zijing Zhang, Geyang Zhang, Jia Min, Qiaoting Shi, Shijie Lyu, Qiuxia Chu, Xingshan Qi, Huimin Li, Yongzhen Huang, Eryao Wang

**Affiliations:** 1Institute of Animal Husbandry, Henan Academy of Agricultural Sciences, Zhengzhou 450002, China; lxmsci@126.com (X.L.); zzh18763091053@163.com (Z.Z.); 18838917969@163.com (X.W.); sqtsw@126.com (Q.S.); sjlyu@outlook.com (S.L.); chuqiuxia@163.com (Q.C.); 2College of Animal Science and Technology, Northwest A&F University, Yangling 712100, China; 3College of Animal Science and Technology, Henan Agricultural University, Zhengzhou 450046, China; 4Bijie Academy of Agricultural Sciences, Bijie 551700, China; liganglu888@aliyun.com; 5Bureau of Animal Husbandry of Biyang County, Biyang, Zhumadian 463700, China; 6Agricultural Comprehensive Administrative Law Enforcement Detachment of Zhengzhou, Zhengzhou 450044, China; lihuimin516@163.com

**Keywords:** LPS, endometritis, inflammatory models, miRNA

## Abstract

Lipopolysaccharide (LPS) is one of the main triggering factors for endometritis in cows. However, the specific pathogenesis of LPS-induced endometritis in dairy cows is not yet fully understood. The aim of this study is to establish an in vitro endometritis model using lipopolysaccharide-induced bovine endometrial epithelial (BEND) cells. BEND cells were treated with different concentrations and durations of LPS. Additionally, miRNA sequencing and target gene prediction were performed on both normal BEND cells and LPS-treated endometrial epithelial cells to screen for differentially expressed microRNAs. This study provides a theoretical basis for further exploring the pathogenesis of endometritis.

## 1. Introduction

The uterus plays a crucial role in the process of embryo attachment and fetal conception. It is comprised of the endometrium, myometrium, and plasma layer. The endometrium is a significant tissue for hormonal action especially and holds a prominent position in reproductive physiology research. The mammalian endometrium is composed of epithelial and lamina propria layers. Normally, endometrial epithelial and stromal cells undergo cyclic changes regulated by ovarian hormones. However, dairy cows who have given birth may experience mechanical damage to their endometrial epithelial cells [1]. During childbirth and calving, the cervix dilates, allowing pathogenic microorganisms from the surrounding environment to enter the birth canal and potentially contaminate the cow uterus [2]. Most bacteria can be eliminated from the body through uterine contractions, while some may become fixed within the uterus. Damage to the shallow endometrial tissue during childbirth can make the uterus susceptible to bacterial infections, leading to the development of endometritis in dairy cows [3]. Prolonged inflammation can result in further damage to the uterine tissue, ultimately impacting the reproductive performance of cows and leading to their premature culling [4].

Endometritis in cows is caused by pathogenic microorganisms, with Gram-negative bacteria such as *Escherichia coli* being the most common culprits [5,6]. The Gram-negative bacteria are recognized by toll-like receptors (TLRs) on the surface of the endometrial cells of the cow, which triggers the congenital immune defense mechanism of the uterus [7,8,9]. The congenital immune defense function of the uterine tissue depends on the endometrial layer of the uterine. Endometrial tissue can express multiple TLRs, immune-related protein, and inflammatory factor [10,11,12], which has an important role in maintaining the stability of the microenvironment in the uterus. Studies have found that lipopolysaccharide (LPS) is the main virulence factor of *E. coli*, which can lead to the acute inflammation process in the animal body [13]. Its pathogenic mechanism is widely studied [14]. The main receptor of LPS is toll-like receptor 4 (TLR4). The interaction of LPS–TLR4 can activate multiple signal channels [15]. These include mitogen-active protein kinases (MAPKS) and nuclear factor-kappa B (NF-κB) signal pathways. These two are classic signal channels related to inflammation [16,17].

Epithelial cells and matrix cells identify LPS through TLR4/CD14/MD2 complexes, thereby activating the inflammatory signal pathway [18]. Subsequently, the activated TLR4 signaling pathway induce the secretion of inflammatory factor, leukocytes, tumor necrosis factor, and prostaglandin. Studies have found that after 24 h of LPS treatment of endometrial epithelial cells and matrix cells, the mRNA expression of *TLR4* signal molecules (TLR4, CD14) increased significantly [19]. After *TLR4* is knocked out, LPS cannot form an effective complex with MD-2 because TLR4 is the receptor it activates. As a result, LPS cannot trigger an immune response, leaving the immune system unable to respond appropriately to bacterial endotoxins [20,21]. Therefore, there is a variety of natural immunities related to the endometrial tissue to identify pathogenic microorganisms and start related signal pathways to fight bacterial infections. However, the specific mechanism by which bacteria cause endometrial tissue infections has not yet been extensively reported.

The microRNA(miRNA) is a single-stranded small molecular non-coding RNA with a length of 21~23 nucleotides (nt) that plays an important role in development, immunity, cardiac function, tumorigenesis, cell proliferation, cell apoptosis, and virus pathology [22,23]. Some miRNAs with deeply conserved sequences showed obvious conserved patterns in their spatial expression, further demonstrating their functional conservatism. For example, miR-124, which is specifically expressed in muscles and is specific to the nervous system, can also be detected in flies, fish, and mammals [24]. After LPS stimulation, the expression of miR-148a in endometrial epithelial cells was significantly downregulated. Overexpression of miR-148a suppressed NF-κB activation by targeting TLR4-mediated signaling pathways, leading to a significant reduction in the production of pro-inflammatory cytokines such as IL-1 and TNF-α [25]. Zhao et al. found that let-7c can reduce the expression of pro-inflammatory cytokines by inhibiting the activation of NF-κB, thereby improving LPS-induced endometritis [26]. Zhao et al. also found that miR-223 can inhibit the activation of the inflammatory mediator NLRP3. This reduces the production of the inflammatory factor IL-1β, thereby protecting uterine tissue from inflammatory damage [27]. Wondim et al. [28] found that compared to the healthy Holstein Frisian cows, miRNA expression analysis revealed a dysregulation of 35 miRNAs, including miR-608, miR-526b*, and miR-1265, in Holstein Frisian cows with clinical endometritis, as well as 102 miRNAs, which include the let-7 family (let-7a, let-7c, let-7d, let-7d*, let-7e, let-7f, let-7i). Additionally, in both Holstein Frisian cows with subclinical and Holstein Frisian cows with clinical endometritis, 14 miRNAs, including let-7e, miR-92b, miR-337-3p, let-7f, and miR-145, were affected. This suggests that changes in miRNA expression profiles have the potential to serve as diagnostic biomarkers for endometritis.

These studies have sparked our interest in miRNA for the treatment of LPS-induced endometritis. In this study, LPS-induced BEND cell inflammation was used to establish a model of endometritis effects of LPS on expression levels of inflammatory factors in BEND cells *IL-1β*, *IL-6*, *IL-8*, *TNF-α* for further analysis. The expression levels of miRNA in different groups were analyzed by high-throughput sequencing. This study provides some data support and ideas for the study of the mechanism of cow endometritis and the development of related therapeutic drugs. Our study also analyzed changes in miRNA expression profiles, which may serve as new biomarkers for endometritis diagnosis.

## 2. Materials and Methods

### 2.1. Cell Culture

The cells used in this trial are BEND cells (American Type Culture Collection (ATCC^®^CRL-2398m, Manassas, VA, USA). The cells were cultured in high-glucose Dulbecco’s modified Eagle’s medium (DMEM, Gibco, Waltham, MA, USA) with 10% fetal bovine serum (FBS, Gibco, CA, USA) and 1% penicillin−streptomycin (HyClone, Logan, UT, USA). All cell lines were cultured in a humidified 5% CO_2_ incubator at 37 °C.

### 2.2. RNA Extraction, RT-PCR, and RT-qPCR

Cell total RNA was extracted by Trizol, and then reverse-transcribed by the reverse transcription kit (AG21102, Accurate Biology, Hangzhou, China). The RNA concentration was detected by multimode reader (SynergyH1, BioTek, Windsor, Vermont, USA). The Evo M-MLV RT kit was used for reverse transcription of miRNA, synthesizing total cDNA. (AG11705, Accurate Biology, Shenzhen, China). The 2 × FastHotstart SYBR qPCR mixture kit (Roche, Mannheim, Germany) was used to conduct real-time quantitative PCR (RT-qPCR). RT-qPCR reactions were performed on the CFX connect real-time system (Bio-Rad, Hercules, CA, USA). All samples were repeated three times, with *β-actin* as an internal control for mRNA expression (Table 1). Relative expression levels were calculated using the comparative cycle threshold (2^−ΔΔCt^) method.

Among the differentially expressed miRNAs, six were randomly selected and verified by RT-qPCR. The miRNAs were extracted using the microRNA rapid extraction kit (Aidlab, RN0501, Beijing, China), and the sequences of each miRNA were downloaded from the miRBase website, and the stem-loop primers and fluorescent quantitative primers were designed, which were synthesized by Tsingke Biotech (Table 2). miRNAs were reverse-transcribed into cDNAs using the stem-loop primers and the reverse-transcription kit (Aidlab, PC6501, Beijing, China). The reverse transcription mixture is shown in Table 3, and the reaction program is shown in Table 4.

### 2.3. Cell Viability Assay

The cell-counting kit-8 (CCK-8) (Beyotime Biotechnology, Nantong, China) was utilized to test cell viability activity. The cell suspension containing 2 × 10^3^ cells (100 µL/well) was seeded into a 96-well plate, placed in the cell incubator and incubated until the cells were plastered. After the cells were completely attached to the wall, the original medium was replaced with LPS medium concentrations of 1 μg/mL, 5 μg/mL, 10 μg/mL, 15 μg/mL, and 20 μg/mL. The blank control (no cells) and negative control (no LPS) groups were set up and incubated for 3 h, 6 h, and 12 h, respectively. The cell culture medium was premixed with CCK-8 solution at a ratio of 9:1. After the LPS treatment, the medium was replaced with 100 µL of the premixed solution, and incubation was continued in the incubator for an additional 3 h. The absorbance was measured at 450 nm with a multifunctional enzyme marker (SpectraMax 190, Sunnyvale, CA, USA).

### 2.4. Flow Cytometry

Detection of the effect of LPS (5 μg/mL) treatment for 6 h on cell apoptosis by flow cytometry: collect 1—5−10^5^ cells and wash the cells twice with pre-cold PBS, centrifuging at 1800 rpm for 5 min at 4 °C each time; add 100 μL of binding buffer (Vazyme, Nanjing, China) and gently mix to obtain a single-cell suspension; add 5 μL of annexin V-FITC (Vazyme, Nanjing, China) and 5 μL of PI staining solution (Vazyme, Nanjing, China), then gently mix; incubate in the dark at room temperature (20–25 °C) for 10 min; add 400 μL of binding buffer (Vazyme, Nanjing, China) and gently mix; after staining, samples should be analyzed by flow cytometry within 1 h. The flow cytometer was set to an excitation wavelength of 488 nm; green fluorescence of FITC was detected in the FL1 channel, while the red fluorescence of PI was detected in the FL2 or FL3 channel, with 10,000 events collected for each sample. Data analysis was performed using FlowJo 10.8.1.

### 2.5. miRNA Data Analysis

The miRNA data analysis software was obtained from ACGT101-miR (LC Sciences, Houston, TX, USA), provided by Lianchuan Biologicals (Hangzhou, China). The analysis process is as follows: (1) Removal of 3’ junction and garbage sequences: obtain clean data; (2) Length screening: sequences with 18–25nt retained base length in plants and 18–26nt retained base length in animals; (3) Various RNA database comparison analyses: compare the remaining sequences to (without miRNA) mRNA, RFam, and Repbase databases with filtering; (4) miRNA identification: obtain valid data and compare precursors and genomes for miRNA identification; (5) miRNA differential analysis; (6) Differential miRNA target gene prediction analysis. The sequencing samples were divided into two groups: the control group was added with LPS 0 μg/mL for 6 h BEND, and the experimental group was added with LPS 5 μg/mL for 6 h BEND. The miRNAs were clustered according to the similarity of the miRNA expression profiles of the samples, indicating that there were differentially expressed miRNAs between the control and experimental groups, compared with the control group. * Shows *p* < 0.05 and was utilized to demonstrate significant differences, ** shows *p* < 0.01 to indicate highly significant differences. For the analysis of samples with biological replicates, the screening of differentially expressed miRNAs was performed with fold_change > 2 or fold_change < 0.5 (|log2FC| > 1) and *p* < 0.05 as the threshold.

### 2.6. Go and KEGG Analysis

MiRNAs are a class of endogenous non-coding RNAs with regulatory functions found in eukaryotes. In animals, when mature miRNAs and AGO proteins form a complex, they can inhibit mRNA translation in the UTR region at the 3’ end; in plants, mature miRNAs form a complex with AGO proteins and bind to target genes. Subsequently, the target gene is sheared at the most intermediate binding site of miRNA, and the mRNA is degraded into fragments and the function of encoded protein is lost. Based on this principle, targetscan 8.0 and miRanda were used to predict the target genes of differentially expressed miRNAs. Subsequently, we conducted the Kyoto Encyclopedia of Genes and Genomes (KEGG) pathway and gene ontology (GO) enrichment analyses using the KOBAS online database. We used screening criteria with *p*-values < 0.05 to identify significantly associated pathways and functional annotations.

### 2.7. Statistical Analysis

SPSS 27.0 was used for two-factor analysis of variance for absorbance values, and one-way analysis of variance between groups and within groups was performed. * means *p* < 0.05, indicating significant difference and statistical significance; ** means *p* < 0.01, indicating that the difference is extremely significant and has statistical significance.

All data were analyzed utilizing GraphPad Prism 6.01 or SPSS 27.0 statistical software, and data were expressed as mean ± standard deviation. The differences between the groups were examined using a two-tailed unpaired T-test or ANOVA.

## 3. Results

### 3.1. In Vitro Endometritis Model

The results showed that both LPS concentration and stimulation time had significant effects on cell survival (Appendix A). There was no significant decrease in cell viability after LPS treatment for 3 h. After 12 h of LPS treatment, cell viability significantly decreased, resulting in an insufficient number of cells for subsequent experiments (Appendix A). Cell viability remained within the optimal range after 6 h of LPS treatment (Figure 1a).

To further verify whether the in vitro endometritis model is successful, the expression of inflammatory factors (*IL-1β*, *TNF*, *IL-6*, *IL-8*) associated with endometritis was studied by RT-qPCR. The results show that after 3 h, 6 h, and 12 h with 5 μg/mL LPS concentration, the expressions of *IL-1β*, *TNF*, *IL-6,* and *IL-8* were significantly increased, indicating that an inflammatory response had occurred in endometrial epithelial cells. Therefore, we selected an LPS treatment time of 6 h and LPS concentration of 5 μg/mL as the model conditions (Figure 1b).

### 3.2. Detection of LPS-Induced Cell Apoptosis by Flow Cytometry

To further validate the usability of the model, we also assessed cellular apoptosis. Compared to the control group, cells treated with LPS (5 μg/mL) for 6 h showed significant apoptosis, but the number of viable cells remains adequate for subsequent experiments. (Figure 2b,c).

### 3.3. MiRNA Length Distribution

Statistical analysis was conducted on miRNA length, and miRNAs with clean reads’ length distribution between 18~26 bp in the samples were selected for length distribution statistics. As shown in Figure 3a,b, the length of the clean reads mainly ranged between 21 and 23 nt and the peak was at 22 nt.

### 3.4. Differential miRNA Expression Analysis

In comparison of LPS-treated group and the control group, a total of 88 differentially expressed miRNAs were screened, of which 37 miRNAs were downregulated and 51 miRNAs were upregulated. (Figure 4a,b).

### 3.5. GO Enrichment and KEGG Pathway Analyses of Differentially Expressed miRNA Target Genes

Among the 88 differentially expressed miRNAs, 21 miRNAs are specific to cattle. Target gene prediction was performed on these 21 differentially expressed miRNAs, resulting in 17,050 target genes (Table 5). Subsequently, GO and KEGG enrichment analyses were conducted on these target genes.

In GO enrichment analysis, results are assigned to Biological Process (BP), Cellular Component (CC), and Molecular Function (MF). Compared to the control group, the differentially expressed miRNAs in the LPS-treated group participate in inflammatory-related biological processes and include negative regulation of cell population proliferation, positive regulation of apoptotic process, positive regulation of cell migration, positive regulation of ERK1 and ERK2 cascade, and negative regulation of cell growth; involved cellular components related to inflammation include Golgi apparatus, endoplasmic reticulum, and mitochondrion; the molecular functions related to inflammation that are involved include protein kinase activity, protein kinase binding, and enzyme binding (Figure 5).

The KEGG enrichment analysis results show that the differentially expressed miRNAs are enriched in inflammation-related signaling pathways, including the MAPK signaling pathway and the TNF signaling pathway (Figure 6a,b). These results lay the foundation for our future research on the role of miRNAs in inflammation.

### 3.6. Target Gene Prediction of Differentially Expressed miRNAs and Verification

Then, to verify the expression results of RNA sequencing, six miRNAs were selected from twenty-one miRNAs belonging to bovines for validation. RT-qPCR was used to validate six miRNAs (miR-155, miR-92a, miR-27a-3p, miR-181a-R-1, miR-let-7b, and miR-151-5p) with significant expression differences (Table 6). The RT-qPCR results were consistent with the sequencing results, thus confirming the miRNA expression pattern (Figure 7).

## 4. Discussion

Endometritis is a superficial inflammation that occurs locally in the endometrial tissue layer [29,30]. This inflammation does not show obvious systemic clinical symptoms [31,32]. The endometrial layer is the primary tissue of the uterus exposed to pathogenic bacteria, while the endometrial epithelium is the first line of defense of the uterus against infection [33]. Toll-like receptors on the surface of endometrial epithelial cells can rapidly recognize pathogenic pattern molecules of pathogenic bacteria, mediate innate immune defense mechanisms in uterine tissues, and secrete acute phase proteins, defensins [34,35,36]. The toll-like receptors can both directly eliminate bacteria and mediate the secretion of inflammatory mediators to recruit immune cells such as neutrophils and lymphocytes to kill the bacteria [37,38]. Endometritis is caused by a variety of bacterial infections, mainly Gram-positive and Gram-negative bacteria, such as *Staphylococcus aureus*, *Escherichia coli,* and *Streptococcus*, among which *E. coli* is the main cause of bacteria. The main component of its cell wall LPS is an endotoxin; when the bacteria die or divide, LPS may be released into the surrounding environment [39]. It causes the activation of the immune system, leading to inflammatory response. Although there are many treatments for endometritis, the prevalence of endometritis is still 20–50% [40]. In addition, the widespread overuse of antibiotics has resulted in antibiotic residues, increased pathogen resistance, and dysregulation of microbiota, which have a great impact on the breeding industry [41]. Therefore, it is of great significance to reveal the mechanism of endometritis and find new therapeutic methods.

To study new methods for treating endometritis, it is a trend to establish a model of endometritis by LPS in vitro. Li et al. [42] established a mouse endometritis model by injecting LPS vaginally, studied the therapeutic effect of resveratrol glycosides on LPS-induced mouse endometritis, and found that resveratrol glycosides significantly alleviated inflammatory cell infiltration in mice with LPS-induced endometritis [42]. Chen et al. [43] established a mouse endometritis model by intrauterine injection of 50 μL of 1 mg/mL LPS, studied the effect of selenium on endometritis, and found that selenium significantly reversed LPS-induced uterine histopathological changes, MPO activity, and inflammatory cytokine levels in vivo. Wu et al. [44] cultured endometrial epithelial (BEND) cells in vitro and treated them with 10 μg/mL LPS to establish an in vitro endometritis model, and found that miR-495 could alleviate bovine endometritis by attenuating the activation of NLRP3 inflammasome. To investigate the therapeutic effects of miRNA on endometritis, we established an in vitro endometritis model by treating BEND cells cultured in vitro with 5 μg/mL LPS for 6 h. Compared to previous studies [45,46], we established an in vitro endometritis model using a lower concentration and a shorter duration of treatment. To the best of our knowledge, this is a novel approach to the study. Several studies have indicated that LPS can induce cell apoptosis [47,48,49], and our results also confirmed this finding, further validating the usability of the model and laying a foundation for our subsequent research on miRNA therapy for endometritis.

With the development of high-throughput sequencing technology, miRNA research tools have become increasingly mature. By comparing the differentially expressed miRNAs in healthy and disease processes and annotating their functions and the signaling pathways involved in regulation through bioinformatics analysis, we aim to provide new clues for disease occurrence, development, diagnosis, and prevention. MiRNA is involved in the regulation of a variety of diseases, and numerous studies have demonstrated the important role of miRNA in dairy cattle diseases [50,51,52,53]. Research has shown that the expression patterns of miRNAs undergo significant changes in mastitis [54]. In a mastitis model infected with Staphylococcus aureus, it was found that the expression of 77 miRNAs was significantly different, with 74 of these miRNAs showing extremely significant differences. These miRNAs influence the development and therapeutic outcomes of mastitis by regulating the inflammatory response and immune modulation processes [55].

MiRNAs can participate in the replication, development, and disease prevention of white blood cells by regulating target genes, playing a crucial role in immune regulation. It is of great significance to inhibit the translation of mRNA by binding to target genes to reduce inflammation and elucidate the mechanism of miRNA in bovine endometritis. Chen et al. [56] demonstrated that miR-196b binds to NRAS mRNA 3 ‘UTR and inhibits the expression of *NRAS*, thereby inhibiting the downstream ERK/NF-κB pathway, resulting in downregulated expression of inflammatory factors. Huang et al. [57] significantly reduced the expression level of MAP38K through overexpression of miR-26a, thereby partially inhibiting the activation of the MAPK signaling pathway and reducing the expression level of inflammatory factors. Liang et al. [58] established a mouse endometritis model with LPS, and alpinetin inhibited the activation of the NF-κB signaling pathway by activating PPAR-γ, thus inhibiting inflammatory response. Our sequencing results showed that differentially expressed miRNAs were mostly enriched in the MAPK signaling pathway, which laid a foundation for subsequent research on the mechanism of endometritis. By performing sequencing analysis on the in vitro endometritis model, we identified eighty-eight differentially expressed miRNAs, of which twenty-one miRNAs belong to cattle. We randomly selected five out of the twenty-one miRNAs for validation, and their expression levels were consistent with the sequencing results, confirming the reliability of the sequencing data. We then conducted GO and KEGG enrichment analyses on these twenty-one differentially expressed miRNAs. The enrichment results showed that the differentially expressed miRNAs were mainly enriched in pathways related to cell proliferation and inflammation, such as the MAPK signaling pathway and the PI3K-AKT pathway. This provides insights for our subsequent research on how miRNAs alleviate endometrial inflammation.

## 5. Conclusions

This experiment stimulated endometrial epithelial cells with varying concentrations of LPS, ultimately establishing the condition for an in vitro endometritis model by exposing endometrial epithelial cells to 5 μg/mL of LPS for 6 h. Sequencing analysis was conducted on the in vitro endometritis model, identifying 21 differently expressed miRNAs related to cattle, which were enriched in pathways associated with inflammation. The reliability of the sequencing results was verified using RT-qPCR. The results of this experiment provide fundamental theoretical evidence for further research on the signaling, metabolic utilization, and immune regulation pathways involved in the development of endometritis in dairy cows, thereby laying the foundation for understanding the pathological mechanisms of endometritis in dairy cattle.

## Figures and Tables

**Figure 1 animals-14-03367-f001:**
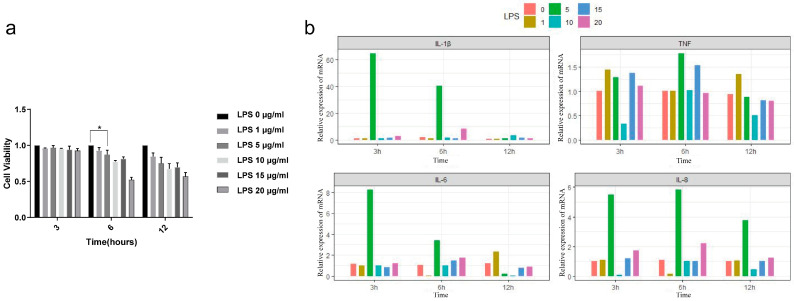
(**a**) Cell activity was measured by CCK-8 after LPS treatment for 3, 6, and 12 h. (**b**) Expression levels of four inflammatory factors, IL-1β, TNF, IL-6, and IL-8, after treatment with 0, 1, 5, 10, 15, 20 μg/mL of LPS for 3, 6, and 12 h. β-actin was used as an endogenous control. Data represent three independent experiments and are presented as the mean ± SEM. * *p* < 0.05.

**Figure 2 animals-14-03367-f002:**
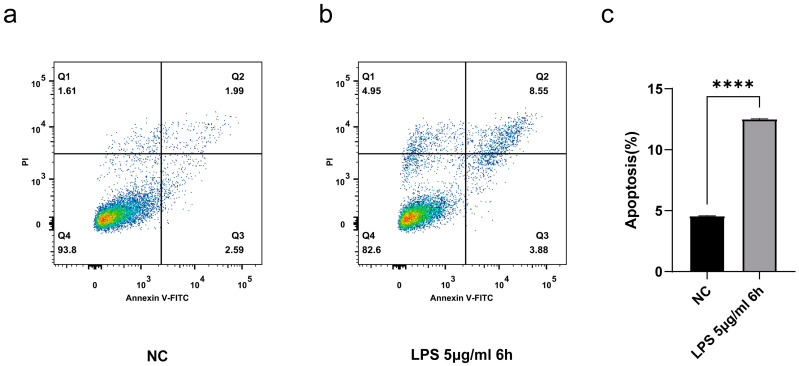
(**a**–**c**) After being treated with LPS 5 μg/mL for 6 h, apoptosis was detected by flow cytometry. β-actin was used as an endogenous control. Data represent three independent experiments and are presented as the mean ± SEM. **** *p* < 0.001.

**Figure 3 animals-14-03367-f003:**
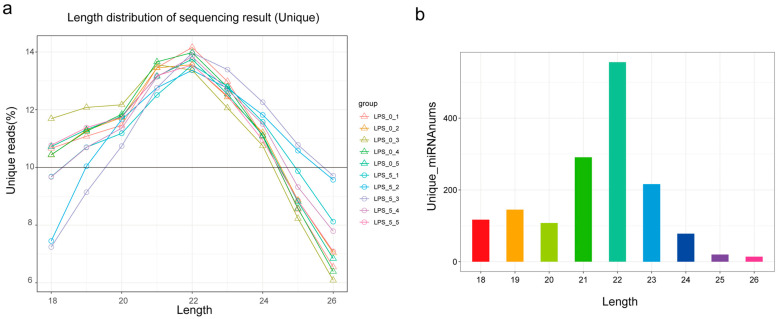
(**a**) Candidate RNA length distribution. (**b**) miRNA length statistics. Note: A statistical analysis of the length of the identified miRNAs was conducted, with the horizontal axis representing the length of the miRNAs and the vertical axis representing the number of miRNAs after deduplication (the reason for deduplication: a precursor of an miRNA aligning to two positions in the genome simultaneously).

**Figure 4 animals-14-03367-f004:**
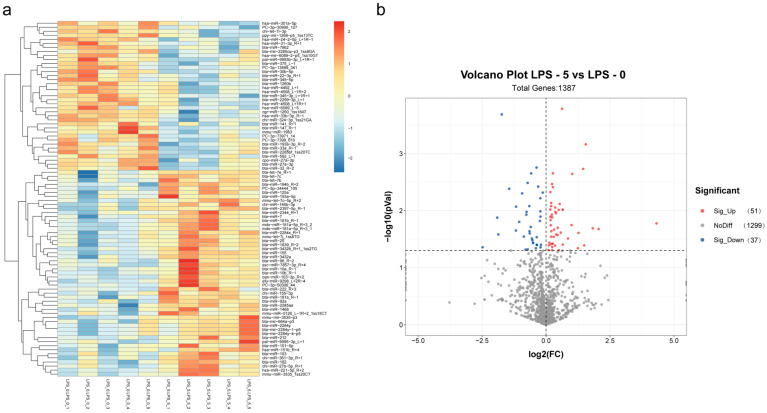
(**a**) Different miRNA cluster heat map. Note: The horizontal coordinates of the heat map are samples (LPS 0-1—LPS 0-5 represent the LPS-untreated group, which is the control group; LPS 5-1—LPS 5-5 represent the LPS-treated group, which is the experimental group), and the vertical coordinates are miRNAs. Different colors represent different miRNA expression levels. Red indicates high-expression miRNA, and dark blue represents low-expression miRNA. (**b**) Different miRNA volcanic map. Note: Each point in the diagram represents an miRNA. The horizontal axis indicates the log2 (fold change) of the expression level difference of miRNA between two samples, while the vertical axis represents the negative log value of the FDR of the miRNA expression changes. A larger absolute value on the horizontal axis indicates a greater fold change in expression levels between the two samples; a larger value on the vertical axis reflects a more significant differential expression. miRNAs with significant differential expression are represented by red and blue points, while those without significant differential expression are shown as gray points.

**Figure 5 animals-14-03367-f005:**
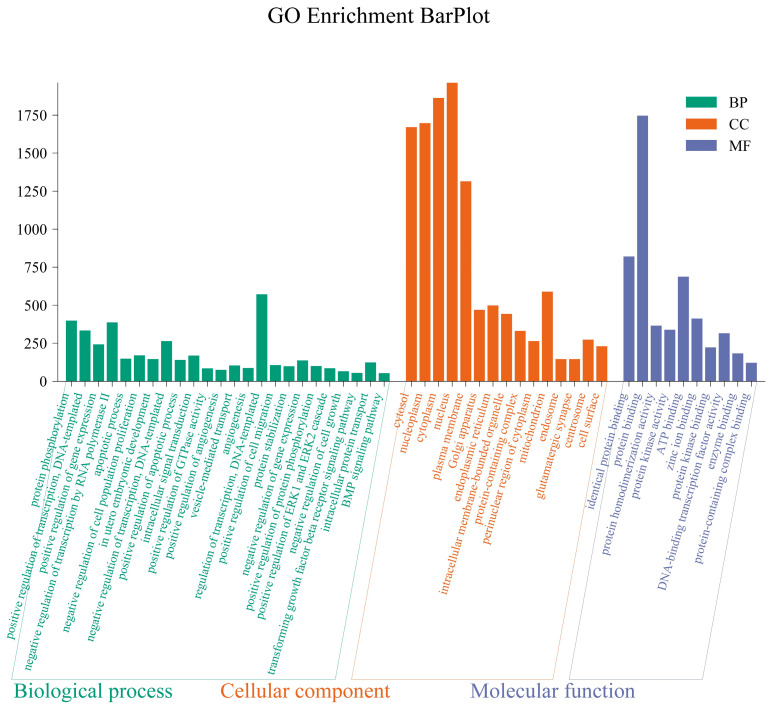
GO enrichment histogram. Note: The horizontal axis represents the GO enrichment terms, and the vertical axis represents the number of target genes enriched under each term. Among all the selected miRNA target genes, the corresponding GO annotations of these target genes were categorized into three classes: Molecular Function, Biological Process, and Cellular Component. The GO enrichment bar chart reflects the distribution of the number of target genes enriched in the terms of BP, CC, and MF. Since there are too many enriched terms in BP, CC, and MF to display all enrichment analysis results, the terms for BP, CC, and MF were sorted from largest to smallest based on the annotated differential gene number (S gene number), and the top 25, top 15, and top 10 terms were selected for graphical representation.

**Figure 6 animals-14-03367-f006:**
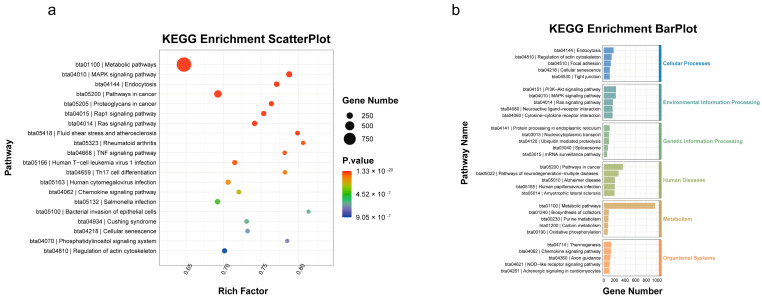
(**a**) KEGG bubble chart. Note: The horizontal axis is labeled as “Ratio”, representing the ratio of the number of differentially expressed miRNA target genes under the specific pathway entry to the total number of target genes. A larger ratio value indicates a higher enrichment of differentially expressed miRNA target genes in that KEGG pathway. The color of the bubble represents the enrichment significance, and the size of the bubble corresponds to the number of genes enriched in that pathway. Bubbles closer to the upper right corner of the diagram indicate higher enrichment and greater relevance, while entries closer to the lower left corner have lower relevance. (**b**) KEGG hierarchy chart. Note: The horizontal axis represents the number of target genes included in the pathway, with colors indicating the primary KEGG classification. The vertical axis represents the names of the pathways.

**Figure 7 animals-14-03367-f007:**
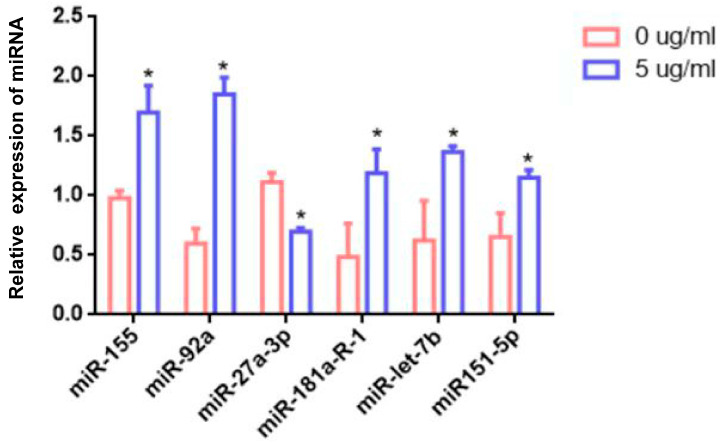
Relative expression of miRNA. U6 was used as an endogenous control. Data represent three independent experiments and are presented as the mean ± SEM. * *p* < 0.05.

**Table 1 animals-14-03367-t001:** Specific primers used for miRNA cDNA generation.

Gene Name	Primer Sequences	Gene Bank ID	Tm (°C)	Length (up)
*β-actin*	F: CATCACCATCGGCAATGAGC	NM_173979.3	60	156
	R: AGCACCGTGTTGGCGTAGAG			
*TNF-α*	F: ACGGTGTGAAGCTGGAAGACAAC	NM_173966.3	60	127
	R:CTGATGGTGTGGGTGAGGAACAAG			
*IL-1β*	F: AACCGAGAAGTGGTGTTCTGCAT	NM_174093.1	60	159
	R: GACTTTGGGGTCTACTTCCTCC			
*IL-6*	F: ATGCTTCCAATCTGGGTTC	NM_173923.2	60	269
	R: TGAGGATAATCTTTGCGTTC			
*IL-8*	F: TTCCACACCTTTCCACCCCA	NM_173925.2	60	126
	R: TCCTTGGGGTTTAGGCAGAC			

**Table 2 animals-14-03367-t002:** Primers used for cDNA verification.

Primer Name	Primer Sequences (5′–3′)
miR-155	RT:GTCGTATCCAGTGCAGGGTCCGAGGTATTCGCACTGGATACGACACCCCT
	F:GCCGCTTAATGCTAATCGTG
	R:GTGCAGGGTCCGAGGT
miR-92a	RT:GTCGTATCCAGTGCAGGGTCCGAGGTATTCGCACTGGATACGACACAGGC
	F:GCCGCTATTGCACTTGTCC
	R:GTGCAGGGTCCGAGGT
miR-27a-3p	RT:GTCGTATCCAGTGCAGGGTCCGAGGTATTCGCACTGGATACGACCGGAAC
	F:GCCGCTTCACAGTGGCT
	R:GTGCAGGGTCCGAGGT
miR-181a-R-1	RT:GTCGTATCCAGTGCAGGGTCCGAGGTATTCGCACTGGATACGACACTCAC
	F:GCCGCAACATTCAACGCTGT
	R:GTGCAGGGTCCGAGGT
miR-let-7b	RT:GTCGTATCCAGTGCAGGGTCCGAGGTATTCGCACTGGATACGACAACCAC
	F:GCCGCTGAGGTAGTAGGTT
	R:GTGCAGGGTCCGAGGT
miR-151-5p	RT:GTCGTATCCAGTGCAGGGTCCGAGGTATTCGCACTGGATACGACACTAGA
	F:GCCGCTCGAGGAGCTCAC
	R:GTGCAGGGTCCGAGGT
U6	F:CTCGCTTCGGCAGCACA
	R:AACGCTTCACGAATTTGCGT

**Table 3 animals-14-03367-t003:** Reverse transcriptional response system.

Components	Volume
Total RNA/miRNA	Up to 2 μg
Seem-loop primer(2μM)	1 μL
5*THERMO Reaction Mix	4 μL
gDNA remover	1 μL
RNase free H2O	To 20 μL

**Table 4 animals-14-03367-t004:** PCR reaction system.

Temperature (°C)	Time	Cycle
25	5 min	1
50	15 min	1
85	5 s	1

**Table 5 animals-14-03367-t005:** The target genes of differentially expressed miRNAs.

miRNA	Gene ID	Symbol	Gene Annotation
bta-miR-155	100313006	USP43	protein deubiquitination
bta-miR-92a	100313392	FRYL	cell morphogenesis
bta-miR-27a-3p	790986	PDP1	cation binding
bta-miR-181a_R-1	100313401	AKAP7	protein-containing complex
bta-let-7b	100170922	PUDP	protein kinase activity
bta-miR-151-5p	790978	TSHZ1	regulation of gene expression

**Table 6 animals-14-03367-t006:** Differentially expressed miRNAs.

miRNA ID	log2FoldChange	*p*-Value	Expression Level
bta-miR-155	0.63	1.63 × 10^−4^	Increased
bta-miR-92a	0.24	3.42 × 10^−3^	Increased
bta-miR-27a-3p	−0.31	3.79 × 10^−3^	Decreased
bta-miR-181a_R-1	0.17	4.73 × 10^−3^	Increased
bta-let-7b	0.23	1.17 × 10^−2^	Increased
bta-miR-151-5p	0.10	4.92 × 10^−2^	Increased

## Data Availability

The sequencing data have been submitted to the NCBI SRA, and are accessible through the accession number PRJNA1018676.

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
