# Peer review of "In Vitro Analysis of LPS-Induced miRNA Differences in Bovine Endometrial Cells and Study of Related Pathways"

_animals, 2024, doi:10.3390/ani14233367_

Round 1
Reviewer 1 Report (Previous Reviewer 1)
Comments and Suggestions for Authors
Dear authors,
I have included revision recommendations in the attached file.
Reviewer.

Must be improved.
Author Response
Dear editor and reviewers,
Thank you for offering us an opportunity to improve the quality of submitted manuscript (animals-3229607). We appreciated very much the reviewers’ constructive and insightful comments. In this revision, we have addressed all of these comments/suggestions. We hope the revised manuscript has now met the publication standard of your journal. We have responded point by point to the comments on the manuscript numbered animals-2995142 submitted to your journal previously.
We highlighted all the revisions in yellow color.
On the next pages, our point-to-point response to the queries raised by the reviewers are listed.
Reviewer #1:
Comment 1: Did you compare the results with a control? after and before?
Response 1: Thanks for your advice. We compare the results with a control after.
Comment 2: Need carity what target gene(s) is/are.
Response 2: Thanks for your advice. We think that simple target gene prediction does not hold specific significance in this article, so the results of this section have been removed. However, since GO and KEGG enrichment analyses need to be based on target genes, this part has been retained in the Materials and Methods section.
Comment 3: Unclear what conditions of cells werestudied for the six miRNA and how the RT.qPCR results agrees with the seg results?
Response 3: Thanks for your advice. We used the same model conditions for treatment and extracted total RNA for RT-qPCR validation.
Comment 4: Is there a specific miRNA for this LPS diagnosis?
Response 4: Thanks for your advice. Currently, there is no specific miRNA to diagnose LPS. However, there are similar examples, such as the clinical diagnostic value of miR-378a-3p in sepsis, which has been validated. Its AUC (Area Under Curve), sensitivity, and specificity were 0.907, 82.5%, and 81.9%, respectively, indicating that it could effectively distinguish sepsis patients from healthy individuals. Based on this research, it can provide experience for miRNAs as diagnostic markers for LPS.
Comment 5: Is this referring to the amplicon size of the genes?
Response 5: Thanks for your advice. This refers to the size of the gene amplification fragment.
Comment 6: Where is the significant difference, standard deviation, mean?
Response 6: Thanks for your advice. Figure 3b shows the length distribution and quantity of significantly differentially expressed miRNAs, without listing the differentially expressed miRNAs.
Comment 7: Not clear what each sample is.
Response 7: Thanks for your advice. This refers to the number of samples submitted for testing, with 5 control groups and 5 LPS-treated groups.
Comment 8: By comparison, is your method better than others? Explain it in your discussion.
Response 8: Thanks for your advice. This refers to the number of samples submitted for testing, with 5 control groups and 5 LPS-treated groups.

Reviewer 2 Report (Previous Reviewer 3)
Comments and Suggestions for Authors
This revised manuscript has been improved. I have no further comments. However, careful proofreading is required before publication. For example, the underline in the title of Table 5 should be removed.
Author Response
Dear editor and reviewers,
Thank you for offering us an opportunity to improve the quality of submitted manuscript (animals-3229607). We appreciated very much the reviewers’ constructive and insightful comments. In this revision, we have addressed all of these comments/suggestions. We hope the revised manuscript has now met the publication standard of your journal. We have responded point by point to the comments on the manuscript numbered animals-2995142 submitted to your journal previously.
We highlighted all the revisions in yellow color.
On the next pages, our point-to-point response to the queries raised by the reviewers are listed.
Reviewer #2:
Comment 1: For example, the underline in the title of Table 5 should be removed..
Response 1: Thanks for your advice. We have made the modifications to this.

Reviewer 3 Report (New Reviewer)
Comments and Suggestions for Authors
This study explored the differential expression of miRNA in vitro endometritis model cells, which is expected to provide a theoretical basis for further investigation of the pathogenesis of endometritis. However, the following problems still need to be further modified:
Abstract
1. Please confirm the full name of CCK-8.
Materials and methods
2. Changes are proposed to the headers of tables 3 and 4.
3. The subheadings of materials and methods are confused.
Results
4. It is recommended to label the abc in the diagram at the top left corner of the diagram.
5. Should the model be built before the subsequent experiments, so should the order of 3.1 and 3.2 of the results be changed?
6. Does the heat map in 3.4 need a simple description?
7. It is recommended to combine 3.6 and 3.7 in one result.
8. The ordinate of RT-qPCR results in figure 7 should be "Relative expression of miRNA".
9. There is a line on the left of the figure in figure 6 a. Please check and update it.
10. The overall discussion of the manuscript results is less, and the previous research results should be discussed first, and then the results will be discussed.
Conclusion
11. The conclusion recommendations are brief and to the point.
Author Response
Dear editor and reviewers,
Thank you for offering us an opportunity to improve the quality of submitted manuscript (animals-3229607). We appreciated very much the reviewers’ constructive and insightful comments. In this revision, we have addressed all of these comments/suggestions. We hope the revised manuscript has now met the publication standard of your journal. We have responded point by point to the comments on the manuscript numbered animals-2995142 submitted to your journal previously.
We highlighted all the revisions in yellow color.
On the next pages, our point-to-point response to the queries raised by the reviewers are listed.
Reviewer #3:
Comment 1: Please confirm the full name of CCK-8.
Response 1: Thanks for your advice. We have completed the name of the CCK-8.
Comment 2: Changes are proposed to the headers of tables 3 and 4.
Response 2: Thanks for your advice. We have revised the titles of Table 3 and Table 4.
Comment 3: The subheadings of materials and methods are confused.
Response 3: Thanks for your advice. We have made changes to the subtitles of the Materials and Methods section.
Comment 4: It is recommended to label the abc in the diagram at the top left corner of the diagram.
Response 4: Thanks for your advice. We have made modifications to the figures.
Comment 5: Should the model be built before the subsequent experiments, so should the order of 3.1 and 3.2 of the results be changed?
Response 5: Thanks for your advice. We have swapped the order of Results 3.1 and 3.2.
Comment 6: Does the heat map in 3.4 need a simple description?
Response 6: Thanks for your advice. We have added a caption to the heatmap in 3.4 for explanation.
Comment 7: It is recommended to combine 3.6 and 3.7 in one result.
Response 7: Thanks for your advice. We have combined 3.6 and 3.7 in one result.
Comment 8: The ordinate of RT-qPCR results in figure 7 should be "Relative expression of miRNA".
Response 8: Thanks for your advice. We have set the vertical axis for the RT-qPCR results in Figure 7 to “Relative expression of miRNA.”
Comment 9: There is a line on the left of the figure in figure 6 a. Please check and update it.
Response 9: Thanks for your advice. We have examined Figure 6a and made the necessary modifications.
Comment 10: The overall discussion of the manuscript results is less, and the previous research results should be discussed first, and then the results will be discussed.
Response 10: Thanks for your advice. We have revised the discussion.
Comment 11: The conclusion recommendations are brief and to the point.
Response 11: Thanks for your advice. We have revised the conclusion.

Round 2
Reviewer 1 Report (Previous Reviewer 1)
Comments and Suggestions for Authors
Authors,
Please review the document and make the necessary changes.
Reviewer.

Author Response
Response to reviewers
Dear editor and reviewers,
Thank you for offering us an opportunity to improve the quality of submitted manuscript (animals-3229607). We appreciated very much the reviewers’ constructive and insightful comments. In this revision, we have addressed all of these comments/suggestions. We hope the revised manuscript has now met the publication standard of your journal. We have responded point by point to the comments on the manuscript numbered animals-2995142 submitted to your journal previously.
We highlighted all the revisions in yellow color.
On the next pages, our point-to-point response to the queries raised by the reviewers are listed.
Reviewer #1:
Comment 1: include this statement only if you did this "relative to the control"
Response 1: Thanks for your advice. We have made alterations to the sentence.
Comment 2: Does the results include the abundance of mRNA? See my previous comment.
Response 2: Thanks for your advice. Yes, our results include the abundance of mRNA. We have made additional supplements to the results section of the article.
Comment 3: Revision needed; full stop?
Response 3: Thanks for your advice. We have made alterations to the sentence.
Comment 4: Please refer to my previous comment; are there any known miRNA used in endometritis diagnosis? If yes, mention here and include reference(s).
Revise the sentence to. ----- a new biomarker for endometritis diagnosis.
Response 4: Thanks for your advice. We have added this part of the research to the article and made modifications to the sentences.
Comment 5: Include the production state.
Response 5: Thanks for your advice. We have made additional supplements for similar places.
Comment 6: Rephrase this sentence to improve readability.
Response 6: Thanks for your advice. We have made alterations to the sentence.
Comment 7: Include the production province.
Response 7: Thanks for your advice. We have made additional supplements for similar places.
Comment 8: Reaction cycle, cycle #, etc. See my previous comment.
Response 8: Thanks for your advice. We have modified this part.
Comment 9: Rewrite these sentences in full sentences and in past tense.
Response 9: Thanks for your advice. We have modified this part.
Comment 10: Didn't you wash the specimen before running it through the machine? See my previous comment.
Response 10: Thanks for your advice. We did not perform washing before loading the samples; we only diluted the samples.
Comment 11: Change to x.
Response 11: Thanks for your advice. We have modified this part.
Comment 12: You need to give explanation to the color boxes. Mention that the expression of the four genes are in response to LPS treatment (include the concentration unit. See my previous comment.
Response 12: Thanks for your advice. We have provided an explanation and specified the concentration units.
Comment 13: change to but.
Response 13: Thanks for your advice. We have modified this part.
Comment 14: Rewrite in full sentence and in past tense.
Response 14: Thanks for your advice. We have modified this part.
Comment 15: Change to sequencing.
Response 15: Thanks for your advice. We have modified this part.
Comment 16: Where is the significant difference, standard deviation, mean?
Response 16: Thanks for your advice. This part is merely a quantitative statistics of miRNA lengths, without significant differences, standard deviations, or means, which we have explained in the figure.
Comment 17: Rewrite: In comparison of LPS treated group and the control group.
Response 17: Thanks for your advice. We have modified this part.
Comment 18: Describe each of the 10 samples' condition.
Response 18: Thanks for your advice. We have described each of the 10 samples' condition.
Comment 19: No corrections were made; see previous comment; do not use et al in this paragraph.
Response 19: Thanks for your advice. We have modified this part.
Comment 20: Place a full stop by it; Figure 5.
Response 20: Thanks for your advice. We have modified this part and have made additional supplements for similar places.
Comment 21: Include, in a separate column in the table, the predicted function information for each gene ID. See previous comment.
Response 21: Thanks for your advice. We have modified this part.
Comment 22: Previous comment unanswered: By comparison, is your method better than others? Explain it in your discussion.
Response 22: Thanks for your advice. We have provided additional explanations for this part in the discussion.
Comment 23: Since you mentioned studies in this sentence, you need to include more than one reference(s).
Response 23: Thanks for your advice. We have added references to support this part.

Round 3
Reviewer 1 Report (Previous Reviewer 1)
Comments and Suggestions for Authors
Authors,
I have included my comments in the attached document.

Author Response
Response to reviewers
Dear editor and reviewers,
Thank you for offering us an opportunity to improve the quality of submitted manuscript (animals-3229607). We appreciated very much the reviewers’ constructive and insightful comments. In this revision, we have addressed all of these comments/suggestions. We hope the revised manuscript has now met the publication standard of your journal.
We highlighted all the revisions in blue color.
On the next pages, our point-to-point response to the queries raised by the reviewers are listed.
Reviewer #1:
Comment 1: Since your mRNA sequencing results include the abundance of mRNA, make sure you mention it in this sentence: eg., the sequencing abundance results.
Response 1: Thanks for your advice. We have added sequencing abundance results.
Comment 2: This sentence need to go to the next paragraph.
Response 2: Thanks for your advice. We have made modifications to this location.
Comment 3: Revise the sentences into one sentence.
Response 3: Thanks for your advice. We have integrated these two sentences into one.
Comment 4: You still have not included reaction cycles in the table.
Response 4: Thanks for your advice. We have added cycles to the table.
Comment 5: According to your response "We did not perform washing before loading the samples; we only diluted the samples". Could you specify in the paragraph the type of staining (i.e., were you staining the cells's exterior components or internal components)?
Response 5: Thanks for your advice. Annexin V-FITC, used to detect membrane externalization, is a marker for early apoptosis, while PI is used to detect changes in nuclear morphology. These staining reagents are also mentioned in the text, indirectly indicating the type of staining.
Comment 6: 1hours change to 1h
Response 6: Thanks for your advice. We have modified this part.
Comment 7: use past tense.
Response 7: Thanks for your advice. We have modified this part.
Comment 8: Revise: 0, 1, 5, 10, 15, 20 ....of LPS.
Response 8: Thanks for your advice. We have modified this part.
Comment 9: Can you include p-value(s)?
Response 9: Thanks for your advice. This section merely provides simple statistics and enumerations, without including p-values.
Comment 10: Coordination changes to coordinates.
Response 10: Thanks for your advice. We have modified this part.
Comment 11: Typo changes to Red.
Response 11: Thanks for your advice. We have modified this part.
Comment 12: (Figure 6a,6b) change to (Figures 6a,6b).
Response 12: Thanks for your advice. We have modified this part.
Comment 13: Has this endometritis model been used before? If no, make sure you mention that to the best of our knowledge, this is a novel approach to the study".
Include references.
Response 13: Thanks for your advice. We have revised this part of the text.

This manuscript is a resubmission of an earlier submission. The following is a list of the peer review reports and author responses from that submission.
Round 1
Reviewer 1 Report
Comments and Suggestions for Authors
Authors,
Please review the attached report.
Reviewer.

Must be improved.
Author Response
Dear Reviewer ,
Thank you for your useful comments and suggestions on the language and the structure of our manuscript. According to your recommendation, we have modified the manuscript accordingly. We hope that this revision could be accepted.
Compared with the manuscript, the revised manuscript had been made some modifications.
We have tried our best to revise and improve the manuscript and made great changes in the manuscript according to the reviewers’ good comments.And here we not only list the changes but marked in red in revised paper.
We appreciate for Editors/Reviewers’ warm work earnestly, and hope that the corrections will meet with approval.
Once again, thank you very much for your comments and suggestions.
Revised by Xinmiao Li , Zijing Zhang Yong-Zhen Huang et al.
May.22,2024

Reviewer 2 Report
Comments and Suggestions for Authors
Line 40 childbirth should be calving
Line 42 cow uterus should be the cows’ uterus
Line 131 140 remove bold
Line 156 – 157 remove: “* Shows P<0.05 and was utilized to demonstrate significant differences, ** shows P<0.01 to indicate highly significant differences”. Add “how you test the prerequisites of the t-test (homogeneity of variances and normal distribution of data”.
· Results section 3.1 you refer to significant differences between 3 levels of the factor time (3h, 6h, and 12h). To find these you need an ANOVA test and not a t-test. You need to review the statistical methodology and amend it.
· “The concentration of the three groups is the highest of the survival rate of endometrial 167 epithelial cells at 5μg/mL. With CCK-8 results, we observed that the cell survival ratio 168 was significantly lower at a treatment time of 6h. Therefore, we chose a treatment time of 169 6h for LPS to further explore the added concentration. The LPS concentration of 5μg/ml 170 stimulated for 6h could therefore be chosen as the model condition”. Here you can only state significant differences between the levels of the factors time and or concentration after an ANOVA. You probably need to also test interactions between levels of the two factors.
· Table 5 in the first column the heading should be time. LPS(ug/mL) should be the heading for 1, 5, 10, 15, 20
· Figure 1 needs to be edited and axis titles should be in English. You also need to explain the acronyms in a note or in the legend and include the units.
· Figure 2a, please edit shortening the y-axis and increasing the font size in the axis titles and legends.
· Edit the title of the 2a figure.
Line 184 26nt ???, also line186 23nt and 22 nt
· Figure 3a is impossible to read the legends. Edit to increase font size or increase figure size. Probably rotating to include in a landscape page format.
· Figure 3b edit legend
· Figure 5b is compressed vertically, making it difficult to read
· Check the impact of any eventual changes in the results section in the discussion and conclusion.
Comments on the Quality of English LanguagePlease amend as indicated.
Author Response

(The authors gave the same response as above.)

Reviewer 3 Report
Comments and Suggestions for Authors
In the present study, the authors examined the impact of LPS on the survival rate and RNA expression of bovine endometrial epithelial cells. The findings provide a foundational theoretical groundwork for subsequent research on the development of endometritis. However, the abstract section lacked clarity and/or detail. The manuscript did not have major concerns. I simply highlighted several comments that could enhance its quality.
General comments
1. In abstract section, what results did you obtain after miRNA sequencing and target gene prediction? Additionally, a concise conclusion should be added to the bottom of abstract section.
2. RT-qPCR or qRT-PCR? Please use “RT-qPCR” as the consistent abbreviation throughout the text. Additionally, please include the annealing temperature, accession number, length, etc., for all primer sequences listed in Table 1. It would be helpful to present this information in tabular form.
3. What is the protocol for dissolving LPS in your study?
4. Could you merge Figures 3 A and B into a single graph?
5. The references are outdated; you should include more recent ones.
Specific comments
1. Line 17, page 1, “Endometritis” should not be bolded.
2. Table 3, “RNase free H2O” should be “RNase free H2O”.
3. Lines 131-140, the sentences should not be bolded.
4. Line 159, the section heading should be 'Results' instead of 'RESULTS'.
5. Line 241, the section heading should be 'Discussion' instead of 'DISCUSSION'.
6. Line 305, the section heading should be '5. Conclusion' instead of '4. Conclusion'.
Author Response

(The authors gave the same response as above.)

Round 2
Reviewer 1 Report
Comments and Suggestions for Authors
Authors,
Please review the current (attached) and recent reports to revise the manuscript accordingly.
Reviewer.

Must be improved.
Reviewer 2 Report
Comments and Suggestions for Authors
There are major statistical issues in your work identified in the first reviewing round and not addressed by the authors. I have to reject the article.